# Efficacy and Safety of COVID-19 Vaccines in Phase III Trials: A Meta-Analysis

**DOI:** 10.3390/vaccines9060582

**Published:** 2021-06-01

**Authors:** Haoyue Cheng, Zhicheng Peng, Wenliang Luo, Shuting Si, Minjia Mo, Haibo Zhou, Xing Xin, Hui Liu, Yunxian Yu

**Affiliations:** 1Department of Public Health and Department of Anesthesiology, The Second Affiliated Hospital of Zhejiang University School of Medicine, Hangzhou 310009, China; 3150101365@zju.edu.cn (H.C.); 22018678@zju.edu.cn (Z.P.); 21918155@zju.edu.cn (W.L.); 21818499@zju.edu.cn (S.S.); minjiamo@zju.edu.cn (M.M.); 11918158@zju.edu.cn (H.Z.); 21818488@zju.edu.cn (X.X.); 2Department of Epidemiology & Health Statistics, School of Public Health, School of Medicine, Zhejiang University, Hangzhou 310058, China; 3Sir Run Run Shaw Hospital, School of Medicine, Zhejiang University, Hangzhou 310016, China; lhui2010@zju.edu.cn

**Keywords:** COVID-19, vaccine, efficacy, safety, meta-analysis

## Abstract

Nowadays, the vaccination with COVID-19 vaccines is being promoted worldwide, professionals and common people are very concerned about the efficacy and safety of COVID-19 vaccines. No published systematic review and meta-analysis has assessed the efficacy and safety of the COVID-19 vaccines based on data from phase III clinical trials. Therefore, this study has estimated the efficacy and safety of COVID-19 vaccines and the differences between vaccine types. PubMed, Embase, the Cochrane Library, CNKI, Wanfang, medRxiv databases and two websites were used to retrieve the studies. Random-effects models were used to estimate the pooled efficacy and safety with risk ratio (RR). A total of eight studies, seven COVID-19 vaccines and 158,204 subjects were included in the meta-analysis. All the vaccines had a good preventive effect on COVID-19 (RR = 0.17, 95% CI: 0.09–0.32), and the mRNA vaccine (RR = 0.05, 95% CI: 0.03–0.09) was the most effective against COVID-19, while the inactivated vaccine (RR = 0.32, 95% CI: 0.19–0.54) was the least. In terms of safety, the risk of overall adverse events showed an increase in the vaccine group after the first (RR = 1.46, 95% CI: 1.03–2.05) or second (RR = 1.52, 95% CI: 1.04–2.20) injection. However, compared with the first injection, the risk of local (RR = 2.64, 95% CI: 1.02–6.83 vs. RR = 2.25, 95% CI: 0.52–9.75) and systemic (RR = 1.33, 95% CI: 1.21–1.46 vs. RR = 1.59, 95% CI: 0.84–3.01) adverse events decreased after the second injection. As for the mRNA vaccine, the risk of overall adverse events increased significantly, compared with the placebo, no matter whether it was the first (RR = 1.83, 95% CI = 1.80–1.86) or the second (RR = 2.16, 95% CI = 2.11–2.20) injection. All the COVID-19 vaccines that have published the data of phase III clinical trials have excellent efficacy, and the risk of adverse events is acceptable. The mRNA vaccines were the most effective against COVID-19, meanwhile the risk and grade of adverse events was minimal, compared to that of severe symptoms induced by COVID-19.

## 1. Introduction

In late 2019, a highly transmissible and pathogenic coronavirus emerged, designated as syndrome coronavirus 2 (SARS-CoV-2) and there followed a pandemic of acute respiratory disease, named “coronavirus disease 2019” (COVID-19) [1,2]. As of 28 May 2021, there were over 168 million confirmed COVID-19 cases and over 3.5 million deaths worldwide, affecting 192 countries and regions (Johns Hopkins University Coronavirus Resource Center, https://coronavirus.jhu.edu/map.html, accessed on 28 May 2021). Vaccines, which can protect people from viral infections in an efficient and sustainable manner [3], play a critical role in human history against infectious diseases. As the COVID-19 pandemic continues to rage, promoting the development of effective vaccines is critical to prevent further morbidity and mortality and, hopefully, limit and even stop the worldwide spread of COVID-19 [4,5].

The rapid increase in morbidity and mortality of the COVID-19 pandemic has led to a drastic shift in the conventional vaccine development paradigm and timelines from a time frame of 10–15 years to 1–2 years [6]. As of 25 May 2021, there are 101 vaccine candidates in clinical development, by use of a broad range of technology platforms from traditional to novel approaches [7,8]. At least 10 vaccines are in phase III clinical trials, 3 of them have ended phase III with positive results [9]. Among them, at least eight COVID-19 vaccines have been granted emergency use and/or full marketing authorization by the regulatory authorities, namely CoronaVac, HB02 (BBIBP-CorV), AZD1222 (ChAdOx1-S), Sputnik V (Gam-COVID-Vac), Ad26.COV2.S, BBV152, BNT162b2 and mRNA-1273 [4,9,10]. Up to now, most countries are promoting vaccination against COVID-19, such as USA, China, India, and Brazil.

To our knowledge, although COVID-19 vaccines have been widely vaccinated all over the world, no comprehensive systematic review and meta-analysis has been published, focusing on the efficacy and safety of them, based on data from phase III clinical trials. Our study will be the first one summarizing the clinical trials of COVID-19 vaccines that are currently widely vaccinated to compare their efficacy and safety, providing useful reference for vaccine selection and promotion.

## 2. Materials and Methods

### 2.1. Search Strategy and Protocol

The protocol was designed according to the Cochrane Handbook [11] and the Preferred Reporting Items for Systematic Reviews and Meta-analyses for Protocols (PRISMA-P) [12]. The search was performed in PubMed, Embase, the Cochrane Library, CNKI, Wanfang and medRxiv to identify all published and prepublication studies. Detailed search strategies for all four databases were provided in the Appendix A. Other sources selected to provide required studies included two vaccine-related websites called “COVID-19 vaccine tracker” (https://vac-lshtm.shinyapps.io/ncov_vaccine_landscape/, accessed on 14 April 2021) and “Up-to-date mapping of COVID-19 treatment and vaccine development” (https://covid19-help.org/, accessed on 14 April 2021). The references of previously published reviews were browsed to check if there were more pertinent clinical trials.

### 2.2. Eligibility Criteria

PICOS (population, intervention, comparator, outcome, and study) design approach was used to defined study eligibility criteria [13]:Population—subjects participated in clinical trials related to COVID-19 vaccines;Intervention—COVID-19 vaccination;Comparator(s)—COVID-19 vaccine or placebo;Outcomes—vaccine efficacy for prevention of COVID-19 (primary outcome) was assessed on the basis of incidences from randomized controlled trials. Secondary outcome was vaccine safety, including adverse events at the injection site (e.g., pain, swelling, induration, erythema), systemic adverse events (e.g., fever, headache, fatigue, muscle pain, joint pain, nausea and/or vomiting, chill) and serious adverse events (e.g., dehydration, syncope, atrial fibrillation, pulmonary embolism, acute kidney injury and so on). During day 0 to 7, 14, or 28 after injections, subjects were asked to record any adverse events.Study designs—randomized controlled trials were eligible for inclusion. Animal studies, case reports, reviews, editorials, letters and conference abstracts were excluded. Articles describing the results of phase III COVID-19 vaccine trials were included first. If there were no safety-related outcomes in the above articles, the results of phase I/II trials of the same vaccine would be included. Studies were excluded if there was an overlap in subjects with another study within the same analysis.

### 2.3. Data Extraction and Quality Assessment

Two authors (Haoyue Cheng and Zhicheng Peng) were independently responsible for data extraction, and disagreements were determined by the third author (Yunxian Yu). For each study, we extracted data on study characteristics (e.g., date of publication, study design, sample size, country, duration of follow-up), population demographics (e.g., ethnicity, sex ratio, mean age, age range) and outcome measures (including COVID-19 incidence in each group and adverse events).

Risk of bias in each study was appraised by pairs of authors using the Cochrane Handbook of Systematic Reviews of Interventions [14]. Only authors experienced in using the tool were involved at this step.

### 2.4. Outcomes

Outcome measures of the meta-analysis included efficacy and safety of COVID-19 vaccines. Vaccine efficacy was evaluated by comparing the difference in the number of COVID-19 cases between the vaccine group and the placebo group after vaccination. The safety outcomes were evaluated through the numbers of adverse events, including local adverse events, systemic adverse events and serious adverse events extracted from the original studies. However, due to the differences in the scope of adverse events in the included clinical trials, the original data of each clinical trial were used without any processing when analyzing the overall adverse events. Meanwhile, the data of the same specific adverse events (e.g., pain at injection site, fever, headache, fatigue, muscle pain, joint pain, nausea or vomiting and chill) in each clinical trial were also extracted for analysis. Besides, because adverse events of HB02 and WIV04 were not separately counted according to the number of injections, these two vaccines lacked the safety outcomes except for serious adverse events.

### 2.5. Data Synthesis and Statistical Analysis

Eight studies were included in the meta-analysis. The magnitude of between-study heterogeneity was estimated using the *I*^2^ statistical parameter, an estimate of the proportion of the total observed variance that is attributed to between-study variance [15]. The differences in incidence of COVID-19 and adverse events with vaccine versus placebo were pooled, stratified across studies or types of vaccines, and all pooled outcome measures were determined using random-effects models with inverse variance weighting.

Pooled effects on vaccine efficacy and adverse events were presented as odds ratio (OR) with corresponding 95% CI. To examine potential publication bias, we produced a funnel plot for outcomes including all studies. All the statistical analyses were conducted using R statistical software VERSION 4.0.0 (R Foundation for Statistical Computing, Vienna, Austria).

## 3. Results

### 3.1. Characteristics of the Studies

A total of 7697 articles from PubMed (807), Embase (1355), the Cochrane Library (301), medRxiv (4920), CNKI (94), Wanfang (102) and other sources (142) were included initially. After screening 3665 titles and abstracts and 37 full text articles, 8 studies [16,17,18,19,20,21,22,23] (one was published on SSRN) providing data on seven COVID-19 vaccines and 158,204 subjects met the eligibility criteria (Figure 1). The eight studies were all randomized, placebo-controlled trials: two mRNA vaccines (mRNA-1273 and BNT162b2), two virus-vectored vaccines (ChAdOx1-S and Gam-COVID-Vac) and three inactivated vaccines (CoronaVac, HB02 and WIV04). The baseline characteristics of the subjects and the design of each study are summarized in Table 1. The vaccination protocols were similar in these studies, including the injection site, number of vaccine injections, the period of follow-up, primary end-point events, and the recorded adverse events. As shown in Figure 2, the risk of bias was low in most studies.

### 3.2. COVID-19 Vaccine Efficacy

For the meta-analysis, the COVID-19 vaccines were divided into three categories according to their types to compare the efficacy based on vaccine versus placebo injection. In general, all the vaccines had good preventive effect on COVID-19 among those who followed the vaccination procedure and received two injections (RR = 0.17, 95% CI: 0.09–0.32) (Figure 3). Besides, the result implied that mRNA vaccine (RR = 0.05, 95% CI: 0.03–0.09) was the most effective against COVID-19, while inactivated vaccine (RR = 0.32, 95% CI: 0.19–0.54) was the least effective among the three types of vaccines.

Taking into account the actual situation, although some subjects only received one injection of vaccination, the meta-analysis finally included the subjects who received at least one dose of the vaccines (Figure 4). Similar to the above result, the vaccines reduced the risk of COVID-19 in the vaccine group relative to the placebo group by 78%. Type-specific efficacy showed that at least one injection of mRNA vaccines (RR = 0.12, 95% CI: 0.05–0.30) was less effective against COVID-19 than two injections of mRNA vaccines (RR = 0.05, 95% CI: 0.03–0.09). However, vaccine efficacy of virus-vectored vaccines and inactivated vaccine was not significantly related to the number of injections.

### 3.3. COVID-19 Vaccine Safety

Generally, total adverse events (RR = 1.46, 95% CI: 1.03–2.05), solicited local adverse events (RR = 2.64, 95% CI: 1.02–6.83) and solicited systemic adverse events (RR = 1.33, 95% CI: 1.21–1.46) showed an increased and statistically significant risk in the vaccine group compared with placebo group after the first injection (Figure 5). Especially, compared with placebo, the risk ratio of local adverse events caused by mRNA vaccine (mRNA-1273) and virus-vectored vaccine (ChAdOx1-S) was 4.26 (4.12–4.40) and 1.62 (1.27–2.05), respectively. It indicated that the risk of local adverse events caused by mRNA vaccine was higher than that of virus-vectored vaccine. Meanwhile, there was no significantly higher risk of systemic adverse events between the vaccine group and the placebo group. In the analysis of specific solicited adverse events, only the risk of pain at the injection site was significantly increased in the vaccine group (RR = 3.67, 95% CI: 1.60–8.43) (Appendix A). Moreover, among all types of vaccines, the risks of all the adverse events induced by the mRNA vaccine were relatively higher, compared with the placebo.

Similarly, after the second injection, total adverse events were significantly higher in the vaccine group than in the placebo group (RR = 1.52, 95% CI: 1.04–2.20) (Figure 6). However, the risk of local (RR = 2.25, 95% CI: 0.52–9.75) or systemic (RR = 1.59, 95% CI: 0.84–3.01) adverse events was not significantly different between the vaccine group and the placebo group. The main reason for this change was that, compared with the first injection, the risk of adverse events among subjects receiving virus-vectored vaccine became lower. The risks of specific adverse events were similar to that of the first injection (Appendix A).

In addition to solicited adverse events, the risk of serious adverse events is also one of the foci of vaccine safety. Compared with the placebo, the COVID-19 vaccines did not increase the risk of serious adverse events (RR = 0.94, 95% CI: 0.71–1.25) (Figure 7). Especially, virus-vectored vaccines reduced the risk of serious adverse events (RR = 0.79, 95% CI: 0.63–0.99).

## 4. Discussion

In this meta-analysis of RCTs, we found that the seven COVID-19 vaccines had excellent efficacy and safety. Among them, the mRNA vaccines were the most effective against COVID-19, while the risk of adverse events was also relatively the highest. Meanwhile, the efficacy of the inactivated vaccines was not as good as the mRNA vaccines, while the safety of them was the best among all types of vaccines.

In fact, all of the COVID-19 vaccines did not increase the risk of serious adverse events. Therefore, there is no firm rule that a certain type of vaccine must be used. Each type of vaccine could be used according to the actual situation of each country and the characteristics of different types of vaccines.

Effective and safe vaccination against COVID-19 is the best strategy to stop viral spread and control the pandemic [24]. After the genetic sequence of SARS-CoV-2 was published, extensive global efforts for COVID-19 vaccine development began. From the traditional platforms, such as inactivated, live attenuated virus, viral protein subunit and replicating or non-replicating viral vectors, to the novel platforms, based on DNA or mRNA, global efforts have brought about the diverse vaccine development platforms [8,24,25]. Our meta-analysis included seven COVID-19 vaccines, namely mRNA-1273, BNT162b2, ChAdOx1-S, Gam-COVID-Vac, CoronaVac, HB02 and WIV04, including both traditional vaccines and novel vaccines. In fact, some vaccines (e.g., BBV152, AD5-nCOV, Ad26.COV2.S) are also undergoing phase III trials and their efficacy has been announced, but the specific data has not been published.

As the results showed, all the COVID-19 vaccines exhibited excellent efficacy, but it was affected by the type of vaccine. After the outbreak of SARS-CoV, the spike (S) protein was quickly identified as the immunodominant antigen of the virus, and the receptor-binding domain of the S1 subunit was the primary target of SARS-COV-2 which can bind and neutralize antibodies [26,27,28]. However, different types of vaccines have different principles for inducing antibodies, which results in the differences in vaccine efficacy [29]. For example, inactivated vaccines stem from virus grown in culture and then chemically inactivated, which may deliver stably expressed, conformationally native antigenic epitopes to induce antibodies [26,30]. Inactivated viruses provide a stable expression of antibodies and can be easily produced in large quantities, so inactivated vaccines are widely used in the prevention of infectious diseases [31]. It is worth noting that after the inactivated virus is injected, there is a certain probability that it cannot induce immune memory [32]. As an emerging type of vaccine, mRNA vaccines have developed rapidly in recent years because of their potency, ability for rapid development, and cost-efficient production [33,34]. After the injection of mRNA vaccine, the host uses the prefusion-stabilized S protein–encoding mRNA to produce the target protein, which induces the immune response [26,35]. Although our study shows that mRNA vaccines had, relatively, the highest risk of adverse events compared to the other types of vaccines, they can be produced rapidly and have potent efficacy, making them ideal for a rapid response to newly emerging pathogens [36]. Consequently, for countries with high transmission rates to stop the spread of COVID-19 in a timely and effective way, mRNA vaccines are a worthy candidate for consideration. Therefore, the efficacy of the inactivated vaccines was the worst in the present study, which can be explained theoretically. Furthermore, the results of the present study showed that regardless of the types of vaccines, the vaccine efficacy in subjects who received two injections was similar to that in subjects who received at least one injection. However, previous meta-analysis indicated that multiple vaccinations of COVID-19 vaccines always showed greater immunogenicity than single vaccinations [15]. The main reason is that subjects who received at least one injection overlapped with those who actually received two injections. However, since the original clinical trials did not provide relevant data, there was no way of distinguishing the subjects according to the number of vaccinations.

Another influencing factor that needs to be considered is the incidence of COVID-19 in different countries. Some clinic trials of different vaccines were conducted in different countries, which would increase the heterogeneity. Prior to widespread vaccination, the cumulative incidence of COVID-19 in Argentina, Brazil, Russia, South Africa, UK, USA and United Arab Emirates was about 38.53, 249.06, 284.29, 579.29, 368.10, 574.28 and 355.82 per 100,000 people, respectively (The cumulative incidence was calculated by using the total number of COVID-19 cases as of June 2020, https://www.worldometers.info/, accessed on 28 May 2021), which meant that the risk of COVID-19 for subjects in different countries was inconsistent. We extracted a portion of the data from the included studies, which were about the results of different vaccines in the same country and similar time period, to compare their efficacy. In Brazil, the efficacy of BNT162b2, and inactivated vaccine CoronaVac was 87.7% (8.1–99.7%) [20] and 50.7% (35.9–62.0%) [22], respectively, which implied that the mRNA vaccine was actually the most efficient.

The results of this study showed that vaccination inevitably induced adverse events, while the risk of serious adverse events was not increased. Besides, the risk of local and systemic adverse events of the second injection was lower than that of the first injection, which may be due to the tolerance of the host. Previous studies also reassured that those serious adverse events, such as immune-mediated adverse reactions, were rare events for most known vaccines [37]. In addition, we concluded that the risks of all the adverse events induced by inactivated vaccine were the lowest. This is because inactivated vaccines cannot replicate in the vaccine recipient, they are thus not capable of causing any significant adverse events, resulting in very few contraindications for their use [32]. Among the mRNA vaccines, few studies discussed the reasons related to the increased risk of adverse events caused by them. Only the COVID-19 vaccine has been approved for marketing, so more studies are still needed to evaluate its safety [38]. Therefore, vaccines should be selected according to personal conditions.

Therefore, the present study summarized the efficacy and safety of the vaccines against COVID-19. However, it has several limitations. First, some studies (phase I/II trials) included insufficient numbers of participants without a broad age range, which indicated there was a potential risk of overestimating or underestimating the vaccine safety. Besides, up to now, most of the clinical trials of the COVID-19 vaccines were conducted in USA, Europe and Latin America, and most of the included subjects were middle-aged. Thus, the results of the trials may differ from the real world. It is necessary to consider more population factors, including adding older adults or children, subjects with special conditions such as pregnancy, and individuals with ethnic and geographical diversity. Second, as some vaccines were not included in the present study, due to a lack of public data, there may exist publication bias. Third, there are several differences in the scope of adverse events in the clinical trials of COVID-19 vaccine. For example, diarrhea is a systemic adverse event in the study of CoronaVac, while it is an unsolicited adverse event in the study of mRNA-1273. Therefore, the risk of adverse events of some vaccines might be underestimated. Fourth, in view of the severity of the epidemic, the follow-up time of all the clinical trials of COVID-19 vaccine is less than half a year, which means the long-term efficacy of the vaccines remain to be verified.

In general, our study is the first meta-analysis summarizing the results of phase III trials in the COVID-19 vaccines. Although the development of COVID-19 vaccines has made great progress, there are still many problems waiting for the efforts of scientists and doctors, such as the efficacy of the vaccines against more transmissible SARS-CoV-2 variants [39,40]. Besides, it is worth noting that the development of vaccines in a very short time necessarily implies that is not yet possible to know their long-term efficacy [9].

## 5. Conclusions

As of now, all the COVID-19 vaccines that have published data of phase III clinical trials have excellent efficacy, and the risk of adverse events is acceptable. The mRNA vaccines were the most effective against COVID-19, while the risk of adverse events induced by them could not be ignored. However, the risk and grade of adverse events induced by the vaccines were minimal compared to those of the severe symptoms induced by COVID-19. We suggest that the type of vaccine could be selected according to the severity of the COVID-19 epidemic and personal conditions.

## Figures and Tables

**Figure 1 vaccines-09-00582-f001:**
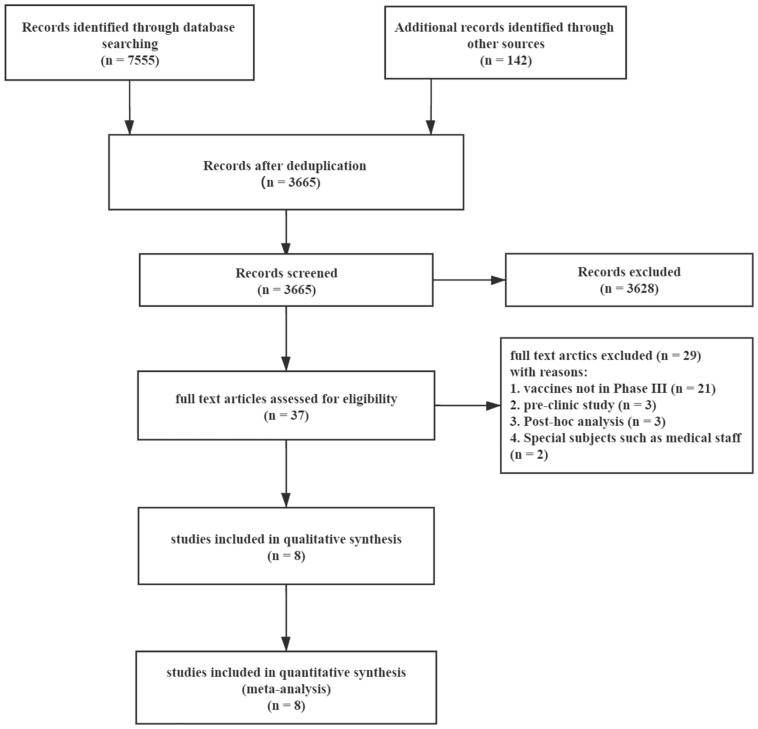
Flow diagram showing the progress through the stages of meta-analysis.

**Figure 2 vaccines-09-00582-f002:**
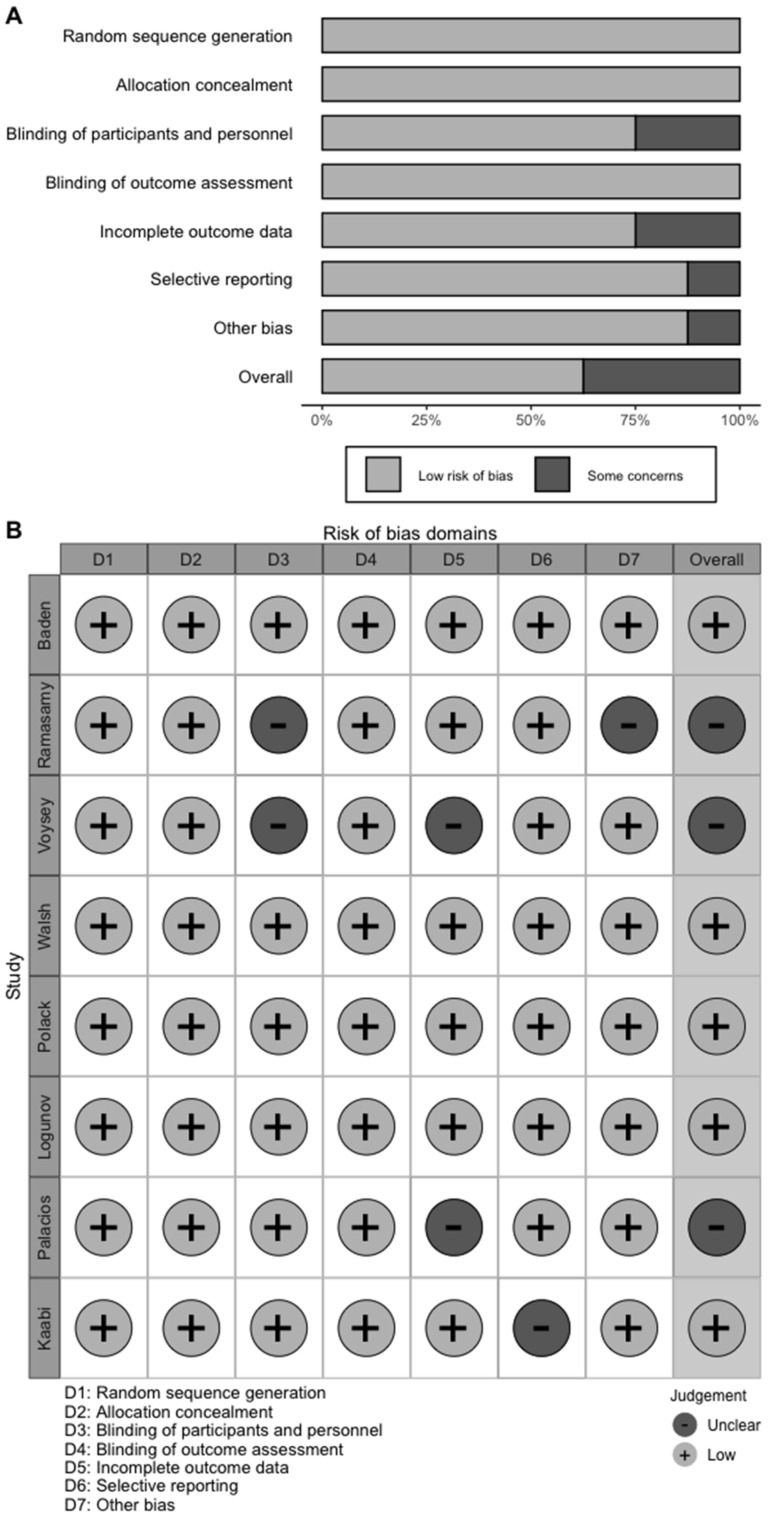
Overall risk of bias assessment using the Cochrane tool and risk of bias assessment by individual trials. (**A**) Overall risk of bias assessment using the Cochrane tool. (**B**) Risk of bias assessment by individual trials.

**Figure 3 vaccines-09-00582-f003:**
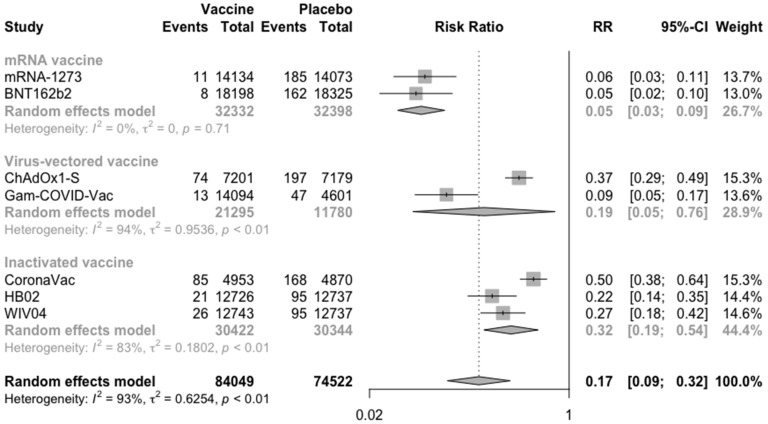
Meta-analysis of efficacy of vaccine between vaccine and placebo groups by type of vaccine (from 7/14 days after dose 2).

**Figure 4 vaccines-09-00582-f004:**
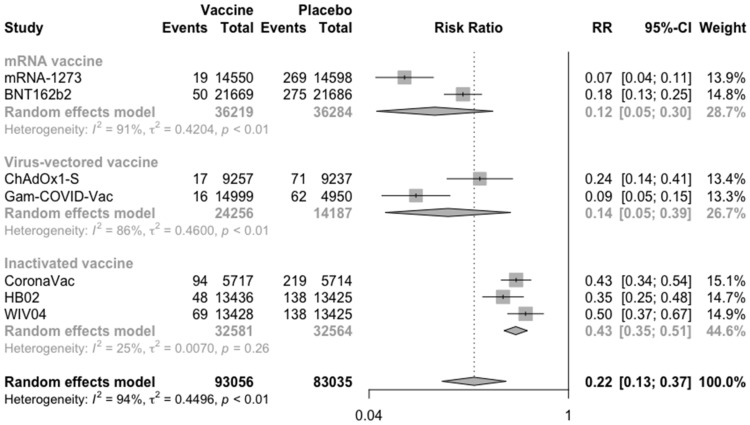
Meta-analysis of efficacy of vaccine between vaccine and placebo groups by type of vaccine (from 7/14 days after dose 1).

**Figure 5 vaccines-09-00582-f005:**
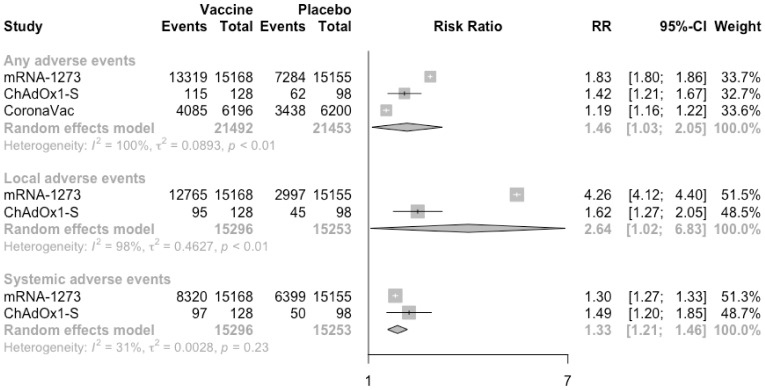
Meta-analysis of effect of vaccine on overall adverse events between vaccine and placebo groups after dose 1.

**Figure 6 vaccines-09-00582-f006:**
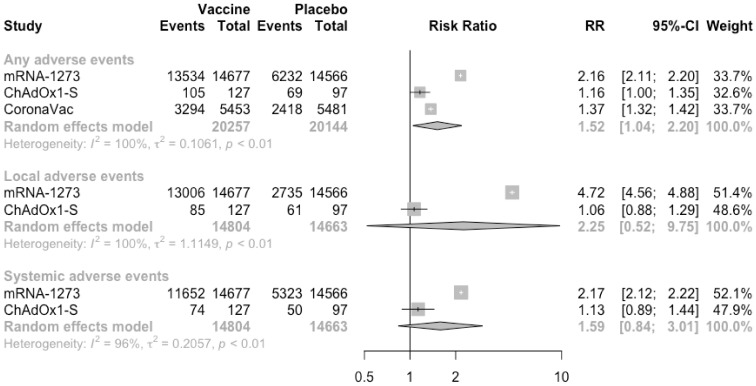
Meta-analysis of effect of vaccine on overall adverse events between vaccine and placebo groups after dose 2.

**Figure 7 vaccines-09-00582-f007:**
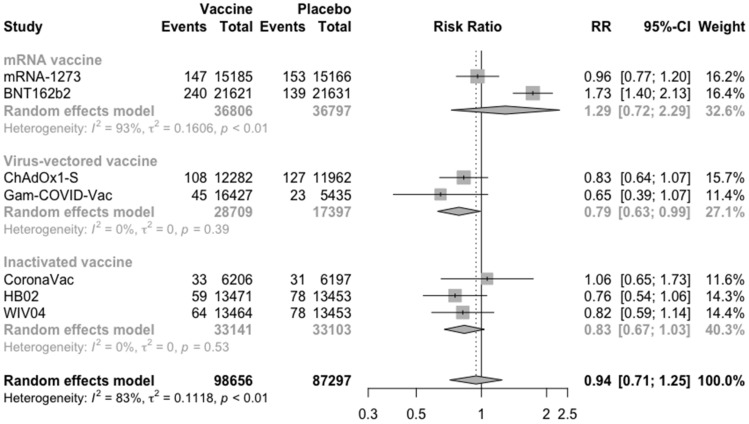
Meta-analysis of effect of vaccine on serious adverse events between vaccine and placebo groups by type of vaccine.

**Table 1 vaccines-09-00582-t001:** Characteristics of the clinical studies included in the network meta-analysis.

Study and Year	Trial Number	Study Characteristics	Vaccine Developer	Dose and Route of Administration	Number of Scheduled Doses (Time of Inoculations)	Type of Candidate Vaccine	Study Duration	Characteristics of Vaccine Recipients	Participating Countries	Age (Mean/Median and Range/SD)	Male (%)
Baden et al., 2021	NCT04470427	Phase III, multicenter, randomized, observer-blind, placebo-controlled	Moderna/National Institute of Allergy and Infectious Diseases’ Vaccine Research Center	mRNA-1273 (100 μg IM)	Prime and boost inoculation (0, 28 days)	mRNA vaccine	27 July 2020 to 21 November 2020	Persons 18 years of age or older with no known history of SARS-CoV-2 infection and with locations or circumstances that put them at an appreciable risk of SARS-CoV-2 infection	USA	51.4 (18–95)	52.7
Ramasamy et al., 2020	NCT04400838	Phase II/III, multicenter, randomized, single-blind, placebo-controlled	University of Oxford/AstraZeneca	ChAdOx1-S (3.5–6.5 × 10^10^ viral particles IM)	Prime and boost inoculation (0, 28 days)	Virus-vectored vaccine	May 30, 2020 to 26 October 2020	Adults aged 18–55 years, then adults aged 56–69 years, and then adults aged 70 years and older, without severe or uncontrolled medical comorbidities	UK	18–55 years: 43.2 (22–55); 56–69 years: 60.4 (56–69.4); ≥70 years: 73 (70–82)	18–55 years: 36.4; 56–69 years: 55.0; ≥70 years: 64.3
Voysey et al., 2021	NCT04324606, NCT04400838, NCT04444674, ISRCTN89951424	Phase II/III, multicenter, randomized, single-blind, placebo-controlled	University of Oxford/AstraZeneca	ChAdOx1-S (3.5–6.5 × 10^10^ viral particles IM)	Prime and boost inoculation (0, 28 days)	Virus-vectored vaccine	23 April 2020 to 6 November 2020	Health adults aged 18 or older	UK Brazil South Africa	≥18	43.6
Walsh et al., 2020	NCT04368728	Phase I, multicenter, randomized, observed-blind, placebo-controlled	BioNTech/Fosun Pharma/Pfizer	BNT162b2 (30 µg IM)	Prime and boost inoculation (0, 21 days)	mRNA vaccine	4 May 2020 to 22 June 2020	Healthy adults 18 to 55 years of age or 65 to 85 years of age	USA	18–55 years: 36.1 (19–54); 65–85 years: 69.1 (65–77)	18–55 years: 38.1; 65–85 years: 47.6
Polack et al., 2020	NCT04368728	Phase III, multicenter, randomized, observed-blind, placebo-controlled	BioNTech/Fosun Pharma/Pfizer	BNT162b2 (30 µg IM)	Prime and boost inoculation (0, 21 days)	mRNA vaccine	27 July 2020 to 14 November 2020	Adults 16 years of age or older who were healthy or had stable chronic medical conditions	USA Argentina Brazil South Africa	52 (16–91)	50.6
Logunov et al., 2021	NCT04530396	Phase III, multicenter, randomized, double-blind, placebo-controlled	Gamaleya Research Institute	Gam-COVID-Vac (1 ± 0.5 × 10^11^ viral particles IM)	Prime and boost inoculation (0, 21 days)	Virus-vectored vaccine	7 September 2020 to 24 November 2020	Adults aged 18 or older with no known history of SARS-CoV-2 infection, and without severe or uncontrolled medical comorbidities	Russia	45.3 (12.0)	61.2
Palacios et al., 2021	NCT04456595	Phase III, multicenter, randomized, double-blind, placebo-controlled	Sinovac	CoronaVac (3 μg IM)	Prime and boost inoculation (0, 14 days)	Inactivated vaccine	21 July 2020 to 16 December 2020	Participants aged 18 or older without previous SARS-CoV-2 infection	Brazil	39.5 (10.8)	35.8
Kaabi et al., 2021	NCT04510207	Phase III, multicenter, randomized, double-blind, placebo-controlled	The Beijing Institute of Biological Products Co, Ltd.,	HB02 (4 μg IM) or WIV04 (5 μg IM)	Prime and boost inoculation (0, 21 days)	Inactivated vaccine	16 July 2020 to 20 December 2020	Participants aged 18 or older without previous SARS-CoV-2 infection	United Arab Emirates (Abu Dhabi Sharjahand Bahrain)	36.1 (9.3)	84.4

## Data Availability

The data presented in this study are available in the article.

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
