# Peer review of "Efficacy and Safety of COVID-19 Vaccines in Phase III Trials: A Meta-Analysis"

_vaccines, 2021, doi:10.3390/vaccines9060582_

Round 1

Reviewer 1 Report

The grammar/sentence structure in both the discussion and the conclusions needs to be significantly improved to make the paper readable.

I think there is an opportunity to strengthen the paper by clarifying in the discussion the pro's and con's of each of the types of vaccine. How much better are the mRNA vaccines at protecting? What does that mean for control in populations with high rates of transmission? What populations would the other vaccines be most beneficial in?

It seems to me that while you report the facts very well, you do not help the reader understand how to use the data that you present. This is a review and therefore should make some broad and specific statements about what the pooled data tells the scientific community. 

Author Response

Comment 1The grammar/sentence structure in both the discussion and the conclusions needs to be significantly improved to make the paper readable.

Response 1Thank you for your advice. We have improved our English usage throughout the manuscript to make it easier to read.

Comment 2I think there is an opportunity to strengthen the paper by clarifying in the discussion the pro's and con's of each of the types of vaccine.

Response 2We have added the advantages and disadvantages of inactivated viruses and mRNA vaccines in the DISCUSSION section and highlight the advantages of the mRNA vaccine.

Comment 3How much better are the mRNA vaccines at protecting? What does that mean for control in populations with high rates of transmission?

Response 3We have compared the efficacy of different types of COVID-19 vaccines in the same country in the DISCUSSION section, which further proved that the efficacy of the mRNA vaccines was the most effective against COVID-19 in our study. And we emphasized their superiority in populations with high rates of transmission in the DISCUSSION section.

Comment 4What populations would the other vaccines be most beneficial in?

Response 4: We didn’t recommend specific vaccines for certain groups of people because the current data in our study can’t support our subgroup analysis of different population. We have suggested that the type of vaccine should be selected according to the severity of the COVID-19 epidemic and personal conditions in each country.

Comment 5: It seems to me that while you report the facts very well, you do not help the reader understand how to use the data that you present. This is a review and therefore should make some broad and specific statements about what the pooled data tells the scientific community.

Response 5: Thank you for your advice. We have deepened and expanded the DISCUSSION section based on your suggestions.

Reviewer 2 Report

The main question is the comparison of different vaccines in efficacy and safety, after extracting data from clinical phase III trials.

Although the topic is not extremely original, the paper provides very useful and important information for the use of the examined vaccines in the short term, based on the available information, hence it adds considerable new knowledge.

The paper is very well and clearly written, with minor english corrections required in the abstract.

The conclusions are consistent with the question posed. However, as mentioned, there is an overstatement about the safety of inactivated vaccines compared to the mRNA vaccines. It should be clearly stated that "systemic" (severe) side effects of all types of vaccines are minimal compared to severe side effects of the disease, and that all available types of vaccines should be used in subjects (with the exception of medical advice otherwise, according to specific pathological conditions) simply based on availability. As the text reads now, it sort of favors indirectly the inactivated vaccines, and this should be corrected and changed.

English language needs minor corrections in the abstract (for example: ...people are concerning..... should be: are concerned about....., etc.

Author Response

Comment 1: The paper is very well and clearly written, with minor English corrections required in the abstract.

Response 1Thank you for your advice. We have corrected the English mistakes in the abstract according to your suggestions.

Comment 2: The conclusions are consistent with the question posed. However, as mentioned, there is an overstatement about the safety of inactivated vaccines compared to the mRNA vaccines. It should be clearly stated that "systemic" (severe) side effects of all types of vaccines are minimal compared to severe side effects of the disease, and that all available types of vaccines should be used in subjects (with the exception of medical advice otherwise, according to specific pathological conditions) simply based on availability. As the text reads now, it sort of favors indirectly the inactivated vaccines, and this should be corrected and changed.

Response 2We have modified the comment on the safety of vaccines and the conclusion has been corrected as follows:

“As of now, all the COVID-19 vaccines that have published the data of phase III clinical trials have excellent efficacy, and the risk of adverse events is acceptable. mRNA vaccines were the most effective against COVID-19, while the risk of adverse events induced by them could not be ignored. However, the risk and grade of adverse events induced by vaccines were minimal compared to that of severe symptoms induced by COVID-19. We suggested that the type of vaccine could be selected according to the severity of the COVID-19 epidemic and personal conditions.”
